# Analysis of the Bioaugmentation Potential of *Pseudomonas putida* OR45a and *Pseudomonas putida* KB3 in the Sequencing Batch Reactors Fed with the Phenolic Landfill Leachate

**Justyna Michalska \*, Artur Piński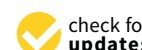, Joanna Żur and Agnieszka Mrozik**

Institute of Biology, Biotechnology and Environmental Protection, Faculty of Natural Sciences, University of Silesia, Jagiellońska 28, 40-032 Katowice, Poland; apinski@us.edu.pl (A.P.); joanna.zur@us.edu.pl (J.Z.); agnieszka.mrozik@us.edu.pl (A.M.)

\* Correspondence: jmichalska@us.edu.pl

**Abstract:** The treatment of landfill leachate could be challenging for the biological wastewater treatment systems due to its high toxicity and the presence of poorly biodegradable contaminants. In this study, the bioaugmentation technology was successfully applied in sequencing batch reactors (SBRs) fed with the phenolic landfill leachate by inoculation of the activated sludge (AS) with two phenol-degrading *Pseudomonas putida* OR45a and *Pseudomonas putida* KB3 strains. According to the results, the SBRs bioaugmented with *Pseudomonas* strains withstood the increasing concentrations of the leachate. This resulted in the higher removal efficiency of the chemical oxygen demand (COD) of 79–86%, ammonia nitrogen of 87–88% and phenolic compounds of 85–96% as compared to 45%, 64%, and 50% for the noninoculated SBR. Simultaneously, the bioaugmentation of the AS allowed to maintain the high enzymatic activity of dehydrogenases, nonspecific esterases, and catalase in this ecosystem, which contributed to the higher functional capacity of indigenous microorganisms than in the noninoculated AS. Herein, the stress level experienced by the microorganisms in the SBRs fed with the leachate computed based on the cellular ATP measurements showed that the abundance of exogenous *Pseudomonas* strains in the bioreactors contributed to the reduction in effluent toxicity, which was reflected by a decrease in the stress biomass index to 32–45% as compared to the nonbioaugmented AS (76%).

**Keywords:** activated sludge; bioaugmentation potential; landfill leachate; phenolic compounds; *Pseudomonas*; wastewater quality

## 1. Introduction

The global upsurge in the urbanization and industrialization is inextricably associated with an increase in municipal and industrial solid waste generation. Although, the European Union Waste Framework Directive (2008/98/EC) has made strict requirements for waste management, landfilling or open dumping still constitute the most prominent practices of waste disposal [1]. The inevitable obstacle arising from waste management is the formation of heavily polluted leachate emerging from the percolation of rainwater through the layers of landfill or dump [2]. Numerous studies have confirmed that the landfill leachate is a reservoir of elevated concentrations of contaminants of emerging concern (CECs), such as phenolic compounds, polycyclic aromatic hydrocarbons, pharmaceuticals, heavy metals, chloride, cyanides, sulfur compounds, humic acids, and ammonia nitrogen, which should be eliminated for the mitigation of leachate toxicity before its discharge into the environment. The landfill leachate composition often varies temporally and spatially, which makes it extremely

difficult to treat [3–7]. So far, a wide range of physical and chemical technologies have been explored for the leachate purification, including coagulation/flocculation, ozonation, membrane filtration, activated carbon adsorption, chemical precipitation, oxidation and ion exchange processes. However, their application in a real scale was reported to be economically unfeasible due to the high operational costs as well as problems concerning the management of chemical sludge and toxic secondary metabolites generated during the treatment process [8–10]. Sewage plants in Poland frequently exploit the biological AS technology for the treatment of the landfill leachate in wastewater stream due to the ease of operation, low financial investment, and environmental friendliness. However, many toxic constituents of the landfill leachate are only partially transformed or not susceptible to degradation in this ecosystem and may hamper the AS process [11,12]. Therefore, the introduction of new metabolic functions into the AS ecosystem by the addition of precisely selected exogenous microorganisms, referred to as bioaugmentation, might provide a solution for the improvement of degradation of recalcitrant and toxic compounds present in the landfill leachate. So far, the bioaugmentation technique has been proved to improve not only the degradation efficiency but also to enhance the bioreactor performance, through the protection of autochthonous microbial communities against adverse effects and compensation for organic or hydraulic overloading [13]. However, the application of this strategy for the treatment of wastewater is not frequently practiced in sewage plants, because the fate of inoculated bacteria is unpredictable and difficult to monitor in the harsh AS conditions [14,15]. Nevertheless, it is assumed that the inoculation of sludge with a group of functionally similar bacterial strains has an advantage over the introduction of a single strain into the bioreactor and may improve the reliability of bioaugmentation, because the mixed microbial consortia are capable of utilizing multiple compounds, due to the mutual complementation of their degradative capacities [16–18].

The successful implementation of bioaugmentation in the AS depends on the efficiency of inoculated bacteria to degrade pollutants under natural conditions and their ability to adapt to the sludge environment and compete with autochthonous bacteria [19]. *Pseudomonas* strains are frequently reported as exhibiting a great metabolic versatility [20,21]. Thus, bacteria belonging to this genus can survive in different ecological niches including environments contaminated with aromatic compounds and they may be considered as potential candidates for the bioaugmentation strategy [22–26]. Unfortunately, information about their use in the bioaugmentation of the biological reactors for the treatment of the industrial and landfill leachates are scarce.

The AS process is usually controlled by the standard operational parameters including the total suspended solids, pH, chemical, and biochemical oxygen demand as well as total phosphorus and nitrogen. However, these parameters show only a slow change over time, which makes it difficult to predict the response of the AS system to any disturbances caused by the leachate and thus preventing the operational problems in the wastewater treatment plant [27]. Therefore, a combination of standard operational parameters with more reliable and accurate methods such as the specific oxygen uptake rate (SOUR), measurement of the ATP concentration, activities of key enzymes involved in the wastewater treatment and microbial community-level physiological profiling can help to determine the specific changes in the AS microbial communities to compare multiple conditions during the biological treatment of the landfill leachate [12].

Herein, the exploitation of bioaugmentation potential of two newly isolated *Pseudomonas putida* OR45a and *Pseudomonas putida* KB3 strains, which harbor the key degradation pathways and have abilities to survive and incorporate into the AS system, was proposed as a desirable approach for the removal of contaminants in wastewater spiked with the phenolic landfill leachate. Given this, individual *Pseudomonas putida* strains and their mixed consortium were inoculated into the SBRs operated in the conditions of long-term exposure to the increasing concentrations of the phenolic landfill leachate. To analyze the effectiveness of bioaugmentation strategy, the specific goals of this research were: (1) evaluating the removal efficiency of the COD, N-NH$_3$ and phenolic compounds in the SBRs, (2) assessing the quality of effluents generated during the leachate wastewater treatment, (3) determining the AS condition, (4) analyzing the functional capacity of indigenous microorganisms,

and (5) establishing the relationships between the factors being studied and evaluating the efficiency of bioaugmentation strategy.

## 2. Materials and Methods

### 2.1. Sample Collection

The landfill leachate was taken from the bottom sediments of the Kalina Pond in Świętochłowice (Upper Silesia, Poland), which serves as the site for uncontrolled and arbitrary disposal of both municipal and industrial waste [12]. The sediment samples were collected in June 2018 at the point of discharge of effluents from the industrial waste dump to the reservoir at a distance of 2 m from the shore and at a depth of 2 m, following the EN ISO 5667-13:2011 [28] and EN ISO 19458:2007P [29].

The AS inoculum was sampled directly from the aeration tank of the wastewater treatment plant Klimzowiec (Chorzów, Poland) and acclimatized to the synthetic wastewater for three weeks before the start of actual bioaugmentation experiment [30].

### 2.2. Bacterial Strains

Two bacterial strains used in this study, *Pseudomonas putida* OR45a (GenBank SPUU00000000.1) and *Pseudomonas putida* KB3 (GenBank SPUT00000000.1), have previously been described as able to degrade high concentrations of phenol and its derivatives [26]. Before the inoculation of the AS, pure colonies of bacterial strains were cultivated separately in the mineral salts medium [26] containing 300 mg/L of phenol (Merck, Darmstadt, Germany) as the sole carbon and energy source at $23 \pm 2\,^{\circ}\mathrm{C}$ with agitation at 130 rpm for 72 h under the proviso that every 24 h a new dose of phenol was applied. Subsequently, the cells were harvested by centrifugation at $4\,^{\circ}\mathrm{C}$ for 15 min at $5.000\times g$ and rinsed three times with sterile mineral salts medium before they were used in the bioaugmentation experiment.

### 2.3. Composition of the Phenolic Landfill Leachate

The concentration of phenolic compounds in the landfill leachate was artificially increased from almost 200 mg/L [12] to 1700 mg/L, to mimic its real composition that was reported in wastewater discharged into the sewage plant Klimzowiec after the revitalization process of the Kalina pond in 2014. For this purpose, the leachate was spiked with a mixture of seven phenolic compounds (Table 1), which occurrence has been previously described in the reservoir [31]. Phenol and its derivatives were acquired from Merck (Darmstadt, Germany).

**Table 1.** Composition of the artificial mixture of phenols in the phenolic landfill leachate.

| Phenolic Compound | Concentration (mg/L) |
|---|---|
| phenol | 327 |
| 3-methylphenol | 356 |
| 4-methylphenol | 356 |
| 3-ethylphenol | 129 |
| 2,4-dimethylphenol | 60 |
| 3,4-dimethylphenol | 281 |
| 2,3,5-trimethylphenol | 292 |

### 2.4. The Experimental Set-Up and Batch Bioaugmentation Experiment

The study was conducted in the experimental model of the SBRs which performs all the functions of the conventional activated sludge plant. The experimental set-up (Figure 1) consisted of eight SBRs: four control bioreactors (C), which were fed only with the synthetic wastewater composed as described in [12] and four bioreactors, which were fed with the synthetic wastewater and three concentrations of the phenolic landfill leachate (L)—3.5% of the influent for the first 32 days of the experiment (I stage), 5.5% of the influent for the next 32 days (II stage) and 12.5% of the influent for another 32 days (III

stage). Two concentrations of the leachate (3.5% and 5.5%) were chosen based on its content reported in the influent discharged into the sewage plant Klimzowiec. In turn, 12.5% of the leachate corresponded with the concentration, which caused almost complete inhibition of the microbial growth in the AS in the acute toxicity testing [12].

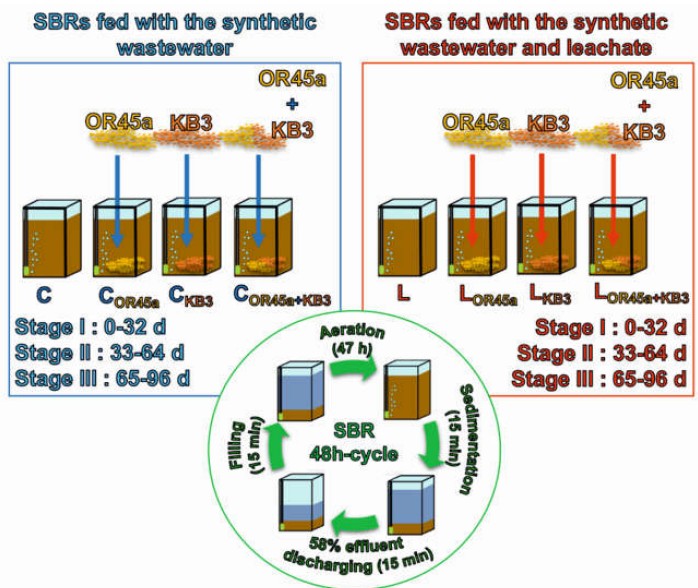

**Figure 1.** Scheme of the experimental set-up.

The AS was inoculated into the bioreactors and mixed with wastewater to obtain a biomass concentration of 3 g/L. Two SBRs: C and L were noninoculated, whereas the bioaugmentation experiment was performed in the bioreactors $C_{OR45a}$ and $L_{OR45a}$ inoculated with *P. putida* OR45a, bioreactors $C_{KB3}$ and $L_{KB3}$ bioaugmented with the *P. putida* KB3, as well as in the bioreactors $C_{OR45a+KB3}$ and $L_{OR45a+KB3}$ inoculated with an equal concentration of biomass of both microorganisms (1:1). The initial amounts of biomass of *P. putida* OR45a and *P. putida* KB3 in the SBRs corresponded to 0.5% of the initial AS biomass (approximately $10^6$ CFU/g of mixed liquor suspended solids, MLSS). To avoid inoculated bacteria being washed out from the AS system, the bioaugmentation procedure was reiterated after the first two times of the removal of effluent.

The dissolved oxygen (DO) level in all SBRs was maintained at a range not exceeding 4 g/L and was monitored using an Elmetron COG-1 oxygen electrode (Elmetron®, Zabrze, Poland) [32]. The pH in the SBRs was not corrected during the experiment. The experiment was conducted within 96 days. After every 32 days and before the addition of a higher concentration of the leachate into the bioreactors, the concentration of the AS biomass in all SBRs was once again brought to 3 g/L by discharging the excess sludge. The bioreactors with a working volume of 9-L operated sequentially at 23 ± 2 °C within a 48-h cycle (Figure 1) [12]. Every second day, 58% of effluent was removed and replaced by the same volume of the synthetic wastewater, with or without the addition of constant portions of the leachate to reflect the operating cycle of the sewage plant Klimzowiec.

*2.5. Determining the Concentration of Selected Phenolic Compounds in Wastewater*

To measure the total concentration of phenol, 3-methylphenol, 4-methylphenol, 2,4-dimethylphenol, 3,4-dimethylphenol, and 2,3,5-trimethylphenol in the SBRs, effluents were sampled after the initial feeding of bioreactors and then every 32 days of their operation. The effluents were centrifuged twice at 4 °C for 15 min at 21,913× *g*. The concentrations of phenolic compounds were determined using the method with 4-aminoantypyrine according to [33]. Because this method does not allow one to measure the concentration of *para*-substituted phenols, the content of phenolic compounds was additionally analyzed chromatographically using a Merck Hitachi HPLC equipped with an Ascentis® Express

C18 HPLC Column (100 × 4.6 mm), an Opti-Solv® EXP precolumn and a DAD detector derived from Merck Hitachi (Darmstadt, Germany). The mobile phase consisted of a mixture of methanol and 1% acetic acid in a ratio of 50:50 (*v/v*) at 1 mL/min flow. The phenolic compounds in the supernatant were identified and quantified by comparing their HPLC retention times and UV-visible spectra with those of the external standards.

## 2.6. Determining the Selected Physicochemical Parameters of the AS and Wastewater Quality

Selected parameters including pH, COD, MLSS, and sludge volume index were measured every 32 days of the SBRs operation as in the reference [32]. The N–NH$_3$ concentration was measured using Nessler's reagent colorimetric method. The turbidity of treated effluent was measured spectrophotometrically at a wavelength of 750 nm using the filtered effluent as a blank to avoid the interference of color generated from the samples [34]. The quality of effluent was expressed as the wastewater quality index calculated using the weighted arithmetic method proposed by [35], which was based on the following parameters of effluent: pH, dissolved oxygen, COD, concentration of phenolic compounds, and N–NH$_3$.

## 2.7. Enumeration of Total Heterotrophic Bacteria in the AS

The heterotrophic microorganisms were extracted from the AS, as described in [12] and enumerated according to [36]. The number of microorganisms able to use phenol as a sole carbon and energy source was calculated by the plate method with mineral salts agar medium supplemented with 300 mg/L of phenol and expressed as the CFU/g of the MLSS.

## 2.8. Measuring the Enzymatic Activity of the AS

Dehydrogenase activity in the AS was determined following the method of [37]. The activity of nonspecific esterase was measured according to [38], whereas the catalase activity was assayed by the method of back-titrating unreacted H$_2$O$_2$ with KMnO$_4$ [39].

## 2.9. Determining the Specific Oxygen Uptake Rate in the Bioreactors

The specific oxygen uptake rate (SOUR) in the SBRs was measured according to [40] with minor modifications. 100 mL of the AS were withdrawn from the bioreactor and introduced to the Winkler bottle placed on a magnetic stir plate. The AS was monitored for the DO depletion using the COG-1 oxygen sensor connected to the DO-meter (Elmetron, Zabrze, Poland). The DO was measured at 30 s intervals until it was completely exhausted. The SOUR in the SBRs were computed according to the following Equation (1):

$$\text{SOUR} \left( \frac{\text{mgO}_2}{\text{g MLSS} \times \text{h}} \right) = \frac{d[\text{DO}]}{dt} \times \frac{1}{\text{MLSS}} \tag{1}$$

where: DO—dissolved oxygen, $\frac{d[\text{DO}]}{dt}$—value of the slope of the DO concentration against time plot, MLSS—mixed liquor solids suspended in the AS.

## 2.10. Determining the ATP Concentration in the AS

Total and dissolved ATP were measured using the BacTiter-Glo$^{\text{TM}}$ Microbial Cell Viability Assay (Promega Corporation, Dübendorf, Switzerland). The cellular ATP was calculated as the subtraction between total and dissolved ATP. Before the measurement of the total ATP, sludge flocs were disaggregated according to [12], which permitted the recovery of bacterial cells from the flocs without affecting the microbial viability. The dissolved ATP was determined in previously filtered samples of the AS with the exclusion of the extraction stage. Both forms of the ATP were extracted from the sludge samples as described in [41]. Subsequently, concentrations of the total ATP and dissolved ATP were determined using the BacTiter-Glo$^{\text{TM}}$ reagent prepared according to the manufacturer's guidelines. For this purpose, 25 µL of the BacTiter-Glo$^{\text{TM}}$ reagent was mixed with the 250 µL of the

sample in the well of Microlite$^{TM}$ Luminescence Microtiter$^{®}$ 96-well plate (Thermo Fisher Scientific, Waltham Massachusetts, USA) and incubated at 37 °C for 20 s. The calibration curve was prepared from a concentration of 10 pM to 1 μM using the ATP solution obtained from dissolving $Na_2ATP \cdot 3H_2O$ in sterilized deionized water (Sigma-Aldrich®, St. Louis, MO, USA).

Based on the concentration of all forms of the ATP in the AS samples, the active biomass and biomass stress indices were calculated according to [42] using the Equations (2) and (3):

$$\text{Active biomass index } (\%) = \frac{\text{dissolved ATP}}{\text{total ATP}} \times 100\% \qquad (2)$$

$$\text{Biomass stress index } (\%) = \frac{\text{cellular ATP} \cdot 0.5}{\text{MLSS}} \times 100\% \qquad (3)$$

where: MLSS—mixed liquor suspended solids in the bioreactors, 0.5—a conversion factor from the ATP concentration to dry biomass concentration.

### 2.11. Assessing the Functional Capacity of Microorganisms in the AS

The community-level physiological profiles using Biolog$^{®}$ 96-well EcoPlates™ (BIOLOG Inc., Hayward, CA, USA) were used to assess the functional capacity of microbial communities in the AS. These analyses were performed at the beginning, after 32 and 64 days of the bioaugmentation experiment. For this purpose, the AS samples collected from the SBRs were prepared according to [12].

The functional capacity indices—Shannon–Weaver diversity index ($H'_{ECO}$), metabolic richness index (S), Gini coefficient (G), average well color development (AWCD) and kinetic parameters of the curve—time at which the total microbial activity increased at the fastest rate ($Avt_{50}$) and maximum microbial activity ($A_{max}$) were estimated by fitting empirical data obtained for each substrate to the Verhulst logistic equation using the Levenberg–Marquardt algorithm implemented in the leastsq function of Python's SciPy package [12] and analyzed using free R-Studio software.

For a more detailed analysis, substrates in the Biolog$^{®}$ EcoPlate$^{TM}$ were divided into seven classes of compounds. In addition, the utilization of carbon sources was expressed as the carbon use index. Concerning overall substrate utilization, the usage of phosphorus and nitrogen-containing compounds was also calculated [43].

### 2.12. Statistical Analysis

All of the data were expressed as the mean and standard deviation and calculated using Microsoft Office Excel 2010. The factors that might influence the changes between the noninoculated and bioaugmented bioreactors during the wastewater treatment were assessed in Statistica$^{®}$ 12.5 PL (TIBCO Software Inc., Palo Alto, CA, USA) using the principal component analysis (PCA) based on a Spearman correlation matrix. The components, which explain the maximum amount of variance in original variables were applied to explore the underlying similarity structures between the SBRs using the cluster analysis. Before clustering, all of the data were first subjected to min–max normalization according to the following Equation (4):

$$ND = \frac{RD_i - RD_{max}}{RD_{max} - RD_{min}} \qquad (4)$$

where, ND (normalized data)—all of the variables scaled to the range (0, 1); $RD_i$—the initial raw data; $RD_{min}$—the minimal value of each parameter in a data set; $RD_{max}$—the maximal value of each parameter in a data set.

The changes between the noninoculated and bioaugmented SBRs during treatment with only the synthetic wastewater or cotreatment with the phenolic landfill leachate were determined using a one-way analysis of variance (ANOVA, post hoc test) and performed in Statistica$^{®}$ 12.5 PL (TIBCO Software Inc., Palo Alto, CA, USA). The analyses allowed to separate the treatments from controls, as

well as among themselves, by applying the post hoc LSD at confidence intervals of 95% ($p < 0.05$). The values that are indicated by different lower case letters were statistically significant.

Bonferroni's test for multiple comparisons was used to study the dissimilarities between the control SBRs and those fed with the leachate and was performed in GraphPad Prism5 (GraphPad Software, San Diego, CA, USA) The differences are indicated by asterisks (Bonferroni correction, $p = \text{ns} \geq 0.05 \geq * > 0.01 \geq ** > 0.001 \geq *** > 0.001 \geq ****$) where ns, *, **, ***, and **** indicate no significant differences, significant differences, very significant differences, and extremely significant differences, respectively.

## 3. Results

*3.1. The impact of Wastewater Composition and Bioaugmentation on the Operational Parameters of the Bioreactors and Effluent Quality*

As shown in Figure 2C,D the increase in the leachate concentration in wastewater was correlated with the gradual decrease in both, the COD and N-NH$_3$ removal efficiency in the bioreactor L. After 96 days of the cotreatment of wastewater with the leachate, the COD and N-NH$_3$ removal efficiency decreased from 95% and 87% to 45% and 64%, respectively. By contrast, bioaugmentation of the SBRs fed with the leachate (L$_{OR45a}$, L$_{KB3}$, and L$_{OR45a+KB3}$) allowed to maintain significantly higher COD and N-NH$_3$ removal efficiency within a range of 79–86% and 87–88%, respectively (Figure 2A,B, Table S1). Interestingly, the appearance of *P. putida* KB3 contributed to the improvement of N–NH$_3$ removal in the control SBRs (C$_{KB3}$ and C$_{OR45a+KB3}$).

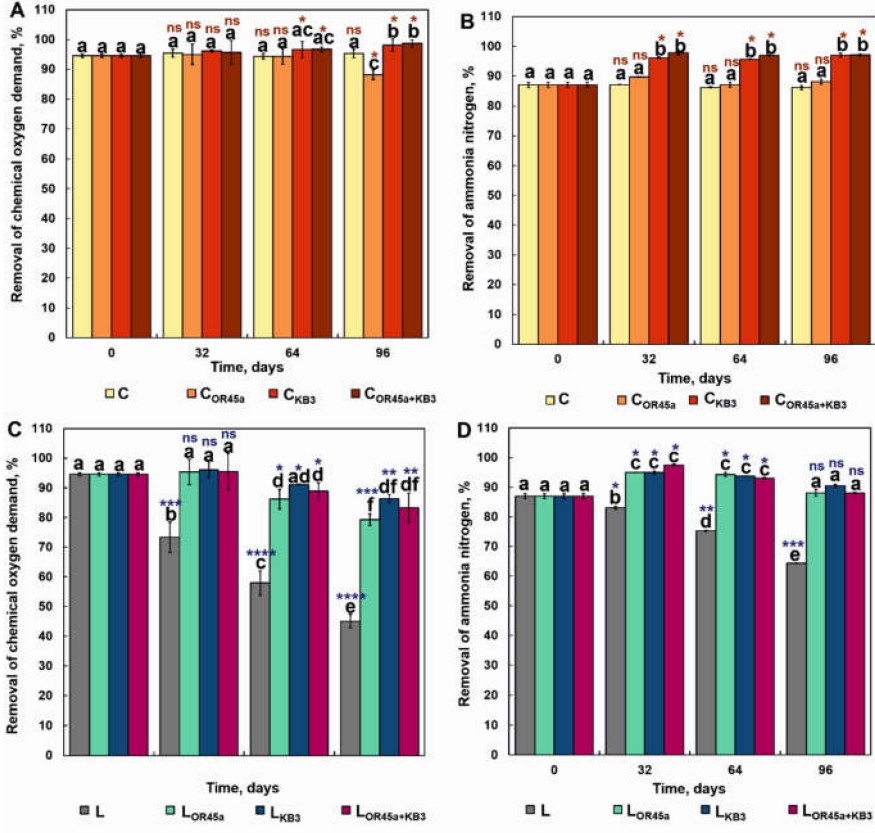

**Figure 2.** Changes in the removal efficiency of chemical oxygen demand and ammonia nitrogen in the noninoculated and bioaugmented activated sludge (AS), fed only with the synthetic wastewater (**A,B**) and cotreated with the phenolic landfill leachate (**C,D**). The treatments are separated among themselves in each stage. The values that are indicated by different lower case letters are statistically significant. The changes in the SBRs during the wastewater treatment are indicated by asterisks.

The removal of contaminants in the bioreactor L was accompanied by the gradual increase in the pH value from 7.22 to 9.47 during 96 days of wastewater treatment (Table S1). The pH in the bioaugmented control SBRs ($C_{OR45a}$, $C_{KB3}$ and $C_{OR45a+KB3}$) showed a generally higher value during the I stage (0–32 day) and III stage (64–96 day) of the experiment. In turn, bioaugmentation of the AS resulted in the production of less alkaline wastewater with the pH value of 7.45, 8.19 and 7.48 in the SBRs cotreated with the leachate ($L_{OR45a}$, $L_{KB3}$ and $L_{OR45a+KB3}$), respectively.

Herein, the efficiency of removal of phenolic compounds in wastewater was higher in the bioaugmented SBRs (Table 2). Microorganisms in the bioreactor L were not able to completely degrade phenolic compounds present in the leachate. Moreover, the increase in the leachate concentration from 3.5% to 5.5% and then to 12.5% resulted in a gradual reduction in removal efficiency of phenolic compounds from 71–76% to 55–58% and then to 45–50%, respectively. The results indicated that enrichment of autochthonous microorganisms with *P. putida* OR45a and *P. putida* KB3 contributed to the complete utilization of phenolic compounds present in wastewater continuously spiked with 3.5% of the leachate within 32 days. In addition, bioaugmentation of the bioreactor with *P. putida* KB3 ($L_{KB3}$) succeeded in the complete removal of phenolic compounds even after the increase of the leachate concentration to 5.5%. However, when the concentration of the leachate was brought to 12.5%, the removal efficiency of phenolic compounds decreased in the bioaugmented SBRs ($L_{OR45a}$, $L_{KB3}$, and $L_{OR45a+KB3}$) by 5–15%.

**Table 2.** Changes in the concentration of phenolic compounds in the noninoculated and bioaugmented sequencing batch reactors (SBRs) fed with the phenolic landfill leachate during 96 days.

| Bioreactor | L | $L_{OR45a}$ | $L_{KB3}$ | $L_{Or45a+KB3}$ |
|---|---|---|---|---|
| Time (days/stage) | Concentration of Phenolic Compounds Determined with 4-Aminoantypyrine (mg/L) | | | |
| 0 (onset of the stage I) | 71.01 ± 3.06 [a] | 72.04 ± 1.95 [a] | 72.64 ± 2.18 [a] | 76.12 ± 4.23 [a] |
| 32 (end of the stage I) | 17.17 ± 2.00 [a] | 0.00 ± 0.00 [b] | 0.00 ± 0.00 [b] | 0.00 ± 0.00 [b] |
| 32 (onset of the stage II) | 126.32 ± 9.05 [a] | 105.55 ± 0.10 [b] | 108.13 ± 1.65 [b] | 111.11 ± 3.18 [b] |
| 64 (end of the stage II) | 53.13 ± 4.65 [a] | 10.77 ± 0.50 [b] | 0.00 ± 0.00 [c] | 3.97 ± 6.02 [d] |
| 64 (onset of the stage III) | 316.19 ± 10.03 [a] | 244.00 ± 3.82 [b] | 233.01 ± 0.12 [c] | 245.50 ± 2.27 [b] |
| 96 (end of the stage III) | 173.80 ± 8.95 [a] | 37.24 ± 5.07 [b] | 10.13 ± 1.03 [c] | 29.04 ± 1.16 [b] |
| Bioreactor | L | $L_{OR45a}$ | $L_{KB3}$ | $L_{Or45a+KB3}$ |
| Time (days/stage) | Concentration of Phenolic Compounds Determined by the HPLC (mg/L) | | | |
| 0 (onset of the stage I) | 52.11 ± 6.64 [a] | 55.06 ± 3.82 [a] | 49.86 ± 1.37 [a] | 52.44 ± 2.28 [a] |
| 32 (end of the stage I) | 14.94 ± 0.53 [a] | 0.00 ± 0.00 [b] | 0.00 ± 0.00 [b] | 0.00 ± 0.00 [b] |
| 32 (onset of the stage II) | 89.22 ± 4.55 [a] | 75.00 ± 0.0.78 [b] | 79.60 ± 0.34 [b] | 74.84 ± 2.34 [b] |
| 64 (end of the stage II) | 39.78 ± 2.22 [a] | 9.66 ± 0.79 [b] | 0.00 ± 0.00 [c] | 5.13 ± 0.21 [d] |
| 64 (onset of the stage III) | 218 ± 5.13 [a] | 189.50 ± 3.60 [b] | 184.97 ± 2.99 [b] | 202.00 ± 8.02 [c] |
| 96 (end of the stage III) | 108.68 ± 9.02 [a] | 19.15 ± 0.39 [b] | 9.55 ± 0.12 [c] | 19.74 ± 7.00 [b] |

The treatments are separated among themselves in each stage. The values that are indicated by different lower case letters are statistically significant.

Knowledge about the AS condition is important for the accurate prediction of the quality of effluent. Herein, the sludge volume index and wastewater turbidity were evaluated to assess the effectiveness of the wastewater treatment process in the SBRs. It was found that the settleability of the AS was negatively affected by the addition of the leachate, which caused a gradual reduction in sludge volume index from 125 cm$^3$/g to 48 cm$^3$/g during the wastewater treatment process in the bioreactor L (Figure 3C). At the same time, the AS in the bioreactor C showed better settleability amounted 103 cm$^3$/g (Figure 3A). Similarly, the inoculated control SBRs underwent some fluctuations in the sludge settleability, achieving the sludge volume index of 142 cm$^3$/g, 108 cm$^3$/g, and 121 cm$^3$/g at the end of the III stage of the wastewater treatment process. Nevertheless, the introduction of *Pseudomonas* strains into the SBRs fed with the leachate led to the improvement of its settleability with the increase in the pollutant concentration from 3.5% to 5.5% (Figure 3C).

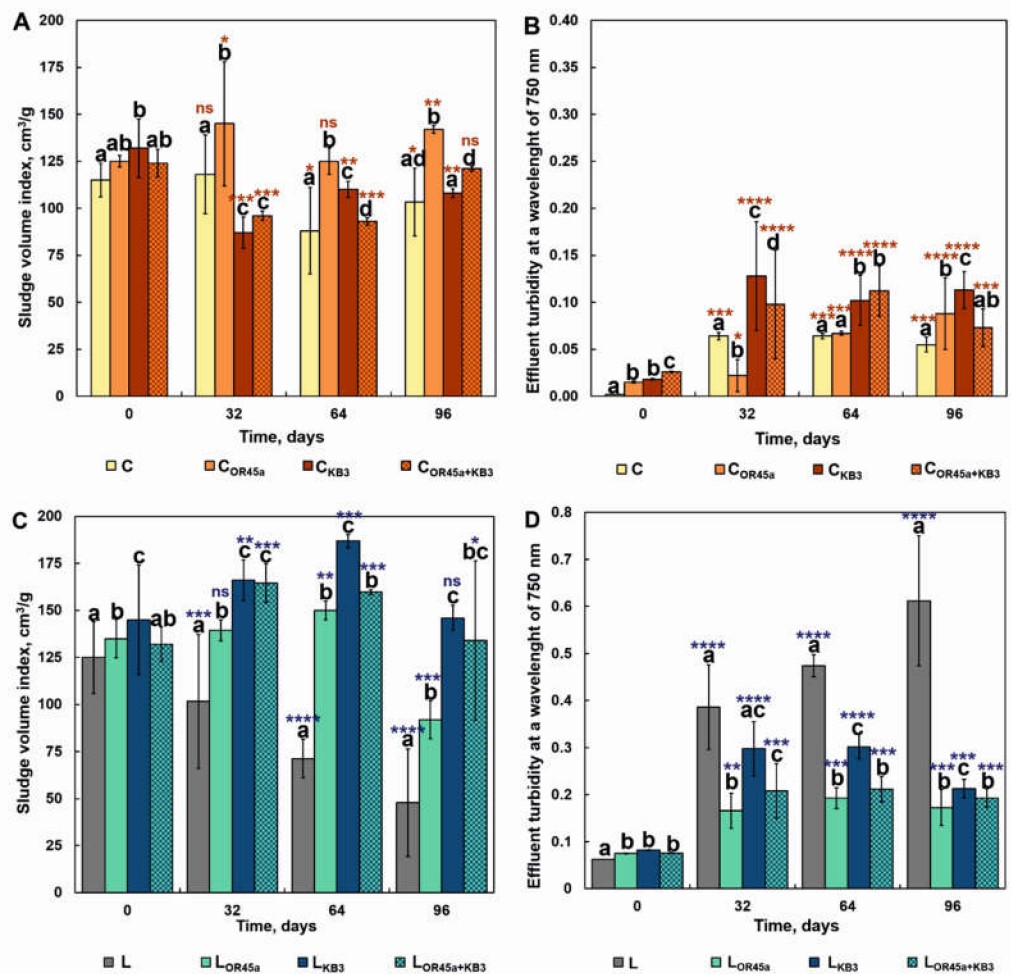

**Figure 3.** Changes in the sludge volume index and wastewater quality in the noninoculated and bioaugmented AS, fed only with the synthetic wastewater (**A**,**B**) and cotreated with the phenolic landfill leachate (**C**,**D**). The treatments are separated among themselves in each stage. The values that are indicated by different lower case letters are statistically significant. The changes in the SBRs during the wastewater treatment are indicated by asterisks.

In this study, the values of effluent turbidity were used to evaluate its quality and the sludge compaction. It was found that the treatment of the leachate-laden wastewater led to the rapid increase in the turbidity of effluent generated in the bioreactor L (Figure 3D). In turn, effluent produced in the bioreactor C was sharply less turbid (Figure 3B). Interestingly, the increase in effluent turbidity was observed during the wastewater treatment in all SBRs inoculated with *Pseudomonas* strains.

Considering the pH value, the concentration of the DO, COD, N-NH$_3$, and phenolic compounds in effluents produced in the SBRs during the treatment of the leachate-laden wastewater (Table S2), it can be concluded that the quality of effluents deteriorated with the increase in the leachate concentration in wastewater. However, significantly lower values of wastewater quality indices were obtained for the bioreactors L$_{OR45a}$ (88.03), L$_{KB3}$ (72.16), and L$_{OR45a+KB3}$ (78.99) than for the bioreactor L (249.76).

*3.2. The Impact of Wastewater Composition and Bioaugmentation on the Biomass Concentration and Number of Heterotrophic Bacteria in the AS*

Through the experiment, the MLSS was found to increase from 3.1 g/L to 4.1 g/L, 4.8 g/L, 4.3 g/L, and 4.7 g/L after 96 days of the wastewater treatment process in the bioreactor C, C$_{OR45a}$, C$_{KB3}$, and C$_{OR45a+KB3}$, respectively (Figure 4A). After the increase in the leachate concentration in wastewater from 3.5% to 5.5% and then to 12.5%, the gradual reduction in the MLSS to 2.4 g/L and then 1.9 g/L was

recorded in the bioreactor L. Similarly, the significant reduction in the MLSS concentration was also observed in the bioreactor $L_{OR45a}$. Contrary, in the bioreactors $L_{KB3}$ and $L_{OR45a+KB3}$ the increase in the leachate concentration was accompanied by the increase in the MLSS value. The changes in total sludge concentration have been reflected in the fluctuations in the number of heterotrophic bacteria in the AS (Figure 4B,D). The presence of phenol-degrading bacteria in the control AS showed a slightly higher abundance of these microorganisms in the bioaugmented SBRs (Figure 4B,D). However, wastewater spiked with the leachate promoted the growth of phenol-utilizing bacteria in all bioaugmented SBRs (Figure 4D).

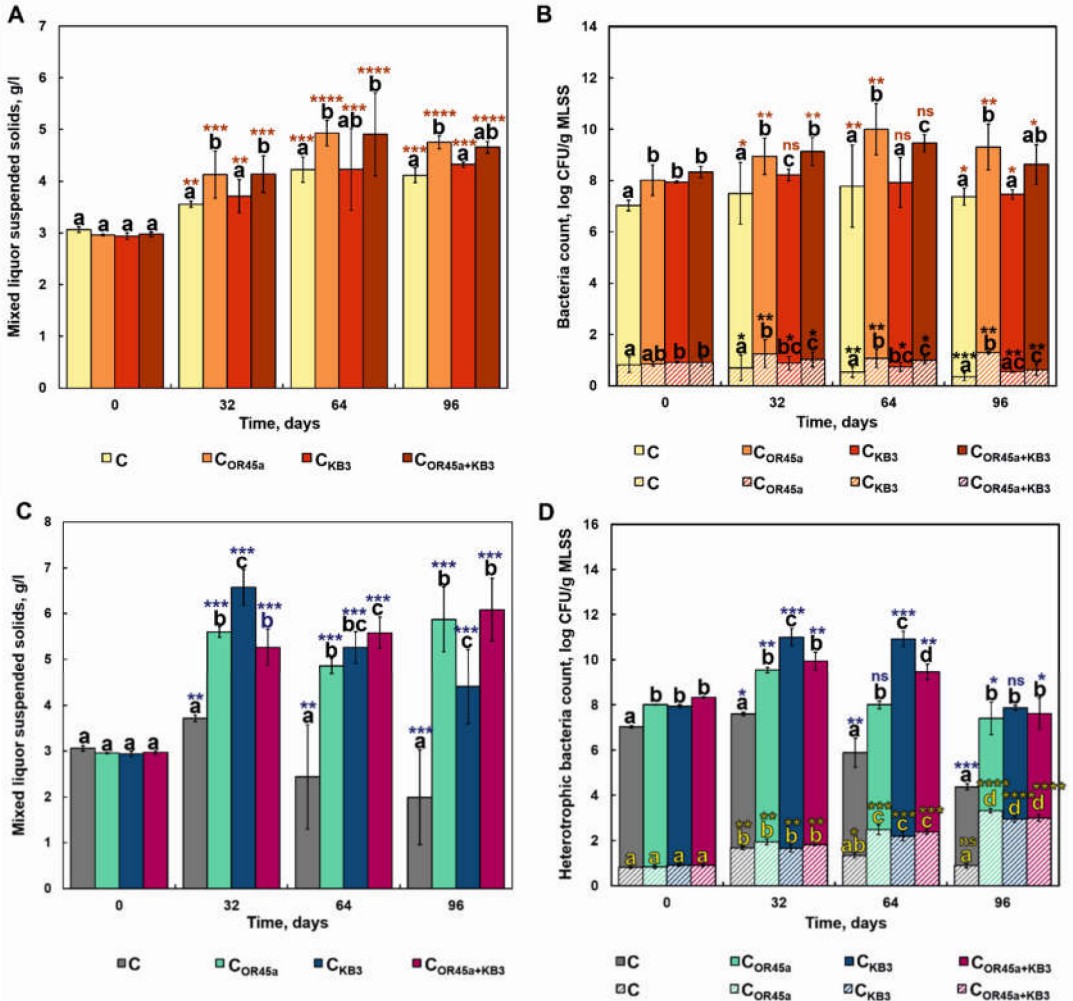

**Figure 4.** Changes in the mixed liquor suspended solids, number of heterotrophic microorganisms (fulfilled bars), and phenol-degrading bacteria (stripped bars) in the noninoculated and bioaugmented AS, fed only with synthetic wastewater (**A**,**B**) and cotreated with phenolic landfill leachate (**C**,**D**). The treatments are separated among themselves in each stage. The values that are indicated by different lower case letters are statistically significant. The changes in SBRs during the wastewater treatment are indicated by asterisks.

Herein, the active biomass index above 55% and stress biomass index below 25% were observed during the synthetic wastewater treatment in the control SBRs (Figure 5A). On the other hand, the constant exposure of the noninoculated AS to the leachate contributed to the decrease in active biomass index by 53%, which was simultaneously correlated with the increase in stress biomass index for this ecosystem to 76%. Inoculation of *Pseudomonas* strains into the SBRs fed with the leachate ensured

active biomass index to be maintained at the level of 47–58% (Figure 5B) and provided a reduction of stress in this ecosystem to 32–45%.

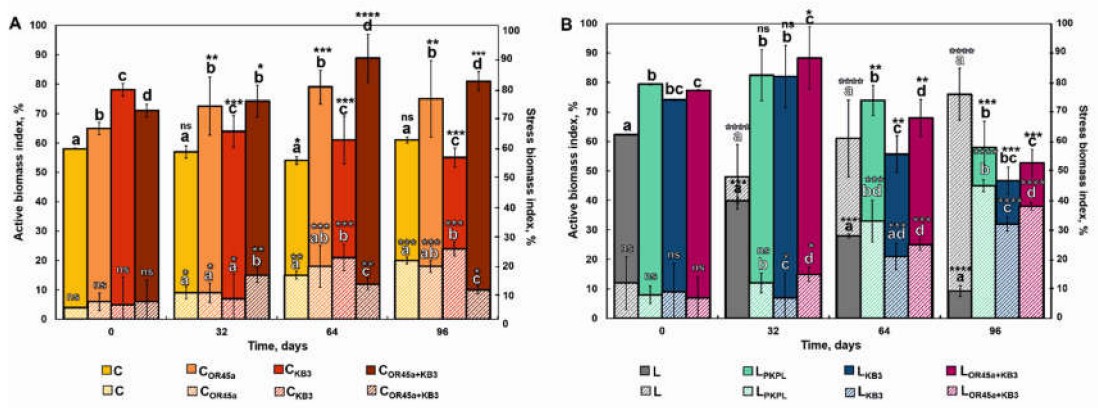

**Figure 5.** Changes in the active biomass index and stress biomass index for the noninoculated and bioaugmented AS fed only with the synthetic wastewater (**A**) and cotreated with the phenolic landfill leachate (**B**). The treatments are separated among themselves in each stage. The values that are indicated by different lower case letters are statistically significant. The changes in SBRs during the wastewater treatment are indicated by asterisks.

### 3.3. The Impact of Wastewater Composition and Bioaugmentation on the Enzymatic Activity of the Bioreactors

The effect of wastewater composition and bioaugmentation on the activities of dehydrogenases, nonspecific esterases, and catalase in SBRs is shown in Figure 6. It was found that the dehydrogenase activity in the bioreactor L was extremely affected by the presence of the leachate and decreased by 76% during 96 days of the wastewater treatment (Figure 6D). The increase in the leachate concentration from 3.5% to 12.5% resulted also in the reduction in respiratory activity in the bioaugmented SBRs in the range from 18% to 27%. By comparison, the dehydrogenase activity in the control SBRs maintained at a constant level (Figure 6A). It was found that the nonspecific esterase activity in the bioreactor L was less sensitive to the leachate (Figure 6E). Nevertheless, the increase in the leachate concentration in wastewater resulted in a rapid decline in the esterase activity by 59%. Simultaneously, the esterase activity in all control SBRs were almost 3–4-times higher (Figure 6B). Regardless of the leachate concentration, the nonspecific esterase activities measured in all SBRs exposed to the leachate were higher than in the bioreactor L. This study also showed that the presence of 3.5% and 5.5% of the leachate in wastewater stimulated the catalase activity in the bioreactor L (Figure 6F). Nevertheless, a significantly greater increase in the catalase activity was observed in the bioaugmented SBRs (Figure 6C,F). The maximum catalase activity was reached after 64 days of wastewater treatment in the bioreactor $L_{OR45a}$. In general, the presence of the leachate in wastewater significantly affected the metabolism of microorganisms in the bioreactor L, which was reflected by the decrease in the SOUR by 53% (Table S1). Conversely, the bacterial respiration rate in the bioreactor C increased during 96 days of wastewater treatment by 11%. The introduction of *Pseudomonas* strains into the AS positively affected the SOUR in all SBRs.

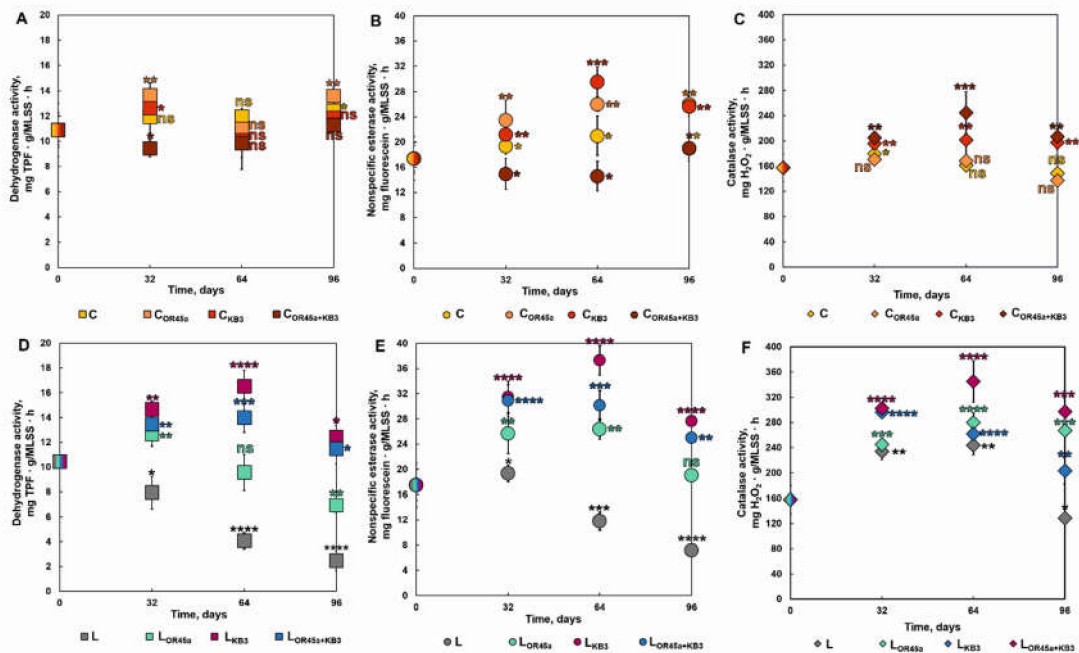

**Figure 6.** Changes in the activity of dehydrogenase, nonspecific esterase, and catalase in the noninoculated and bioaugmented AS fed only with the synthetic wastewater (**A–C**) and cotreated with the phenolic landfill leachate (**D–F**). The changes in SBRs during the wastewater treatment are indicated by asterisks.

*3.4. The Impact of Wastewater Composition and Bioaugmentation on the Functional Capacity of Microorganisms in the AS*

To compare the microbial activity in the SBRs, the EcoPlate$^{TM}$ functional capacity indices and kinetic parameters were computed at the beginning (0d) and after 32, 64, and 96 days of the experiment (Table S3). It was found that the constant feeding of the AS with the increasing concentration of the leachate resulted in a decrease of the Shannon–Weaver diversity index for the bioreactor L as compared to its value for the bioreactor C. In turn, the continuous exposure of the bioaugmented SBRs to the leachate affected the microbial functional capacity to a lesser extent.

The decrease in the microbial functional capacity in the SBRs fed with the leachate was correlated with the decrease in maximum microbial activity in the AS. Once again, the lowest maximal activity ($A_{max}$) was recorded for the bioreactor L (0.37), whereas the microbial activities in the bioaugmented SBRs were higher with a value of 0.90, 0.93, and 1.03 for the $L_{OR45a}$, $L_{KB3}$, and $L_{OR45a+KB3}$, respectively.

The continuous cotreatment of the leachate-laden wastewater in the noninoculated SBRs strongly affected the ability of the AS microorganisms to utilize carbon sources on the Biolog® EcoPlate$^{TM}$ (Tables S3 and S4). After the exposure of the AS to 12.5% of the leachate, microorganisms were able to use only 33.87% among the available substrates (Table S4). It was found that the bioaugmentation of the SBRs fed with the leachate allowed to maintain the high utilization efficiency of the EcoPlate$^{TM}$ compounds at the level of 67.74%, 73.13%, and 80.65%, in the bioreactors $L_{OR45a}$, $L_{KB3,}$ and $L_{OR45a+KB3,}$ respectively. Moreover, the addition of *Pseudomonas* strains into the control SBRs increased the utilization efficiency of carbon sources to 100%.

It was shown that the AS in the SBRs fed with the leachate achieved the highest utilization efficiency of the nitrogen-containing substrates of 41.50% (Table S4). Instead, the ability of the autochthonous microorganisms to use nitrogen compounds did not change significantly in the control SBRs. Interestingly, the cotreatment of wastewater with the leachate contributed to the gradual decrease in the utilization efficiency of the phosphorus compounds. It is worth emphasizing that the utilization of phosphorus sources in the AS bioaugmented with *P. putida* OR45a increased along with the increase

of the leachate concentration. The opposite phenomenon was observed in the SBRs inoculated with *P. putida* KB3 and mixed microbial consortium.

The pattern of different carbon sources utilized by the AS microbiome during the synthetic wastewater treatment and cotreatment with the leachate is presented in Figure 7. Compared to the bioreactor C, the presence of the leachate in wastewater primarily affected the utilization efficiency of carbohydrates, amino acids, surfactants and polymers in the bioreactor L (Figure 7A,B). After 96 days, a sharp decrease in the intensity of the usage of carbohydrates from 25.30% to 1.74% and rise in the intensity of the amino acids and surfactants usage from 15.66% to 43.51% and from 9.50% to 18.82% was recorded in the bioreactor L, respectively (Figure 7B). Additionally, microorganisms in this bioreactor could not use the polymers as carbon sources. Contrary, bacteria in the bioreactors L$_{OR45a}$, L$_{KB3}$, and L$_{OR45a+KB3}$ were characterized by the higher utilization of carboxylic acids of 25–35%, phenolic acids of 5–10% and polymers of 2–3%.

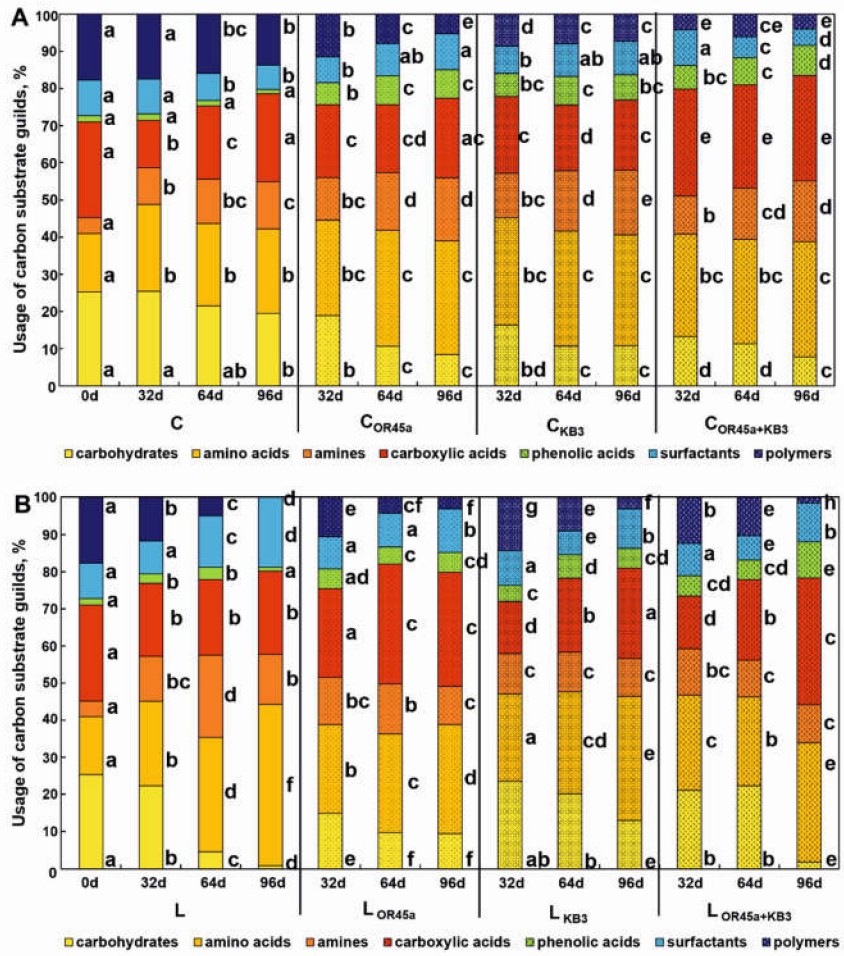

**Figure 7.** Changes in the utilization of carbon sources in the noninoculated and bioaugmented AS, fed only with the synthetic wastewater (**A**) and cotreated with the phenolic landfill leachate (**B**). The treatments are separated among themselves and the changes in the SBRs during the wastewater treatment are indicated by different lower case letters.

## 3.5. Bioaugmentation Potential Analysis of P. putida OR45a, P. putida KB3, and their Consortium

The effectiveness of bioaugmentation strategy was evaluated separately based on the operational parameters of wastewater treatment, the quality of the AS and generated effluent, enzymatic activity and functional capacity indices (Figure 8). The cluster analysis indicated that the bioaugmentation of the AS with *P. putida* OR45a contributed in general to the improvement of the respiratory activity of

microorganisms, ammonia removal efficiency, and settling abilities of the sludge (Figure 8A). In turn, inoculation of the bioreactor with *P. putida* KB3 stimulated the microbial enzymatic activity, MLSS, functional capacity, removal of contaminants as well as the utilization of carbon and phosphorus sources. The introduction of both *Pseudomonas* strains into the SBRs also resulted in the improvement of the AS respiratory and catalase activity, MLSS, and functional capacity. Instead, it did not contribute to the increase in the removal efficiency of contaminants in the AS but led to a reduction in the biomass stress index.

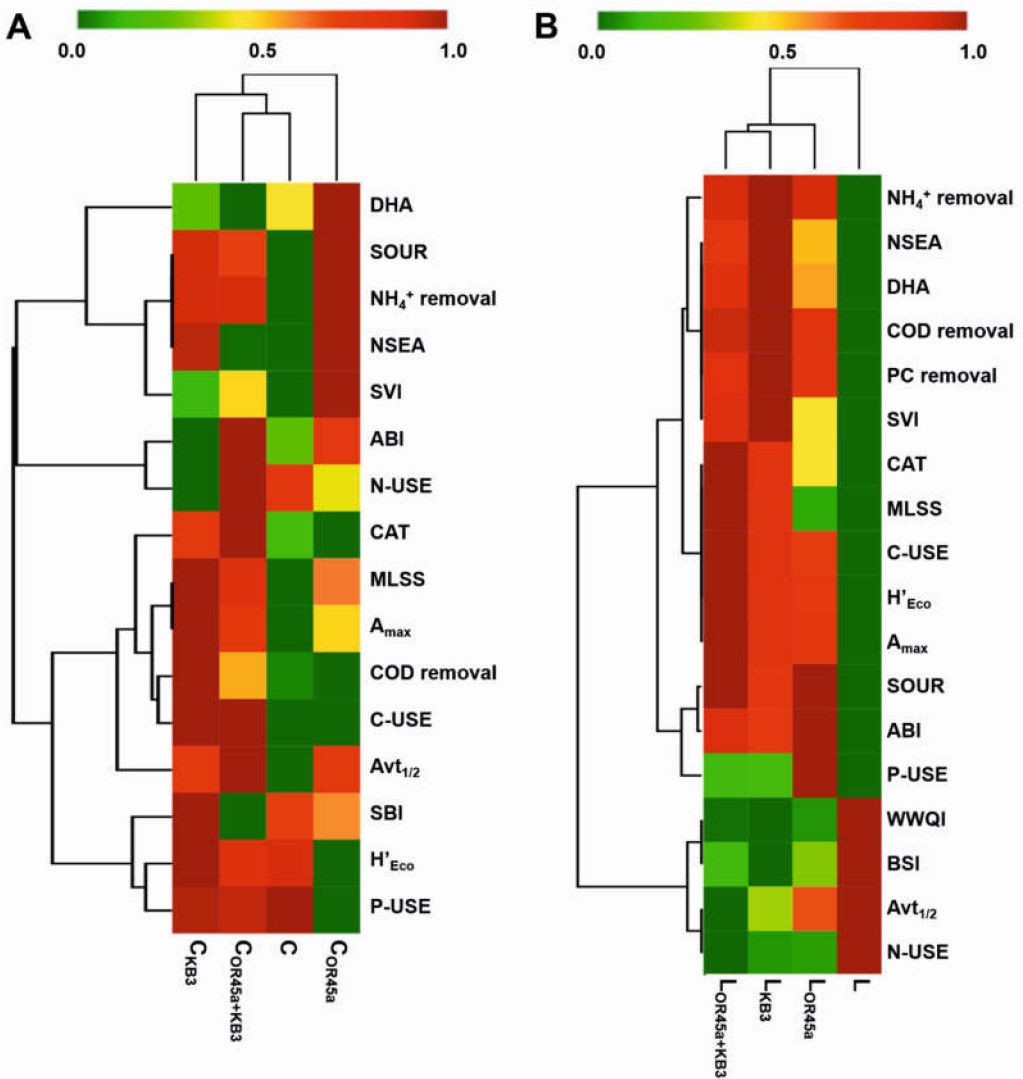

**Figure 8.** Heat maps and cluster analyses based on significant Spearman's rank correlation coefficients calculated for the operational parameters of the SBRs, functional capacity indices, microbial biomass, and activity determined after 96 days in the noninoculated and bioaugmented AS, fed only with the synthetic wastewater (**A**) and cotreated with the phenolic landfill leachate (**B**).

Analysis of the conditions in the AS fed with the leachate after 96 days showed a strong correlation between the bioaugmentation and improvement of the leachate-laden wastewater treatment (Figure 8B). However, the inoculation of the AS with *P. putida* KB3 and the mixed microbial consortium was slightly more effective than the bioaugmentation with *P. putida* OR45a, which led to the significant improvement of the enzymatic activity in this ecosystem and contributed to the generation of better-quality effluents.

## 4. Discussion

The controlled and proper treatment of the landfill leachate are indispensable to minimize the toxicity of this effluent before its discharge into the environment. To ensure the preservation of the public health and ecosystem, the remediation and disposal of the landfill leachate should proceed in an environmentally safe approach. Although both physical and chemical methods have been successfully applied to treat the landfill leachate within a short duration, these techniques are usually expensive and may lead to the unexpected secondary pollution of the environment [8–10]. Therefore, the biological methods are widely considered to be more appropriate for long term treatment of the landfill leachate [44]. However, biological processes often deal with system instability and deterioration of its performance under different stresses [13]. Although the bioaugmentation has proven to be a valuable technique to improve the operation performance and recovery in the biological systems for wastewater treatment, studies that decipher the impact of this strategy on the landfill leachate treatment in the SBRs are scarce.

In the previous study, we described *P. putida* OR45a and *P. putida* KB3 strains as suitable candidates for the bioremediation of wastewater spiked with the landfill leachate [26]. Herein, these bacterial strains introduced into the SBRs fed with the phenolic landfill leachate significantly improved the removal efficiency of contaminants by over 30%. This corresponded simultaneously with the generation of good-quality effluent with a quality index of 57–88 and slightly alkaline pH. On the other hand, the gradual decrease in the removal efficiency of the COD from 95% to 45% in noninoculated bioreactor exposed to the leachate resulted in the formation of very poor-quality effluent with a quality index of 250 and strongly alkaline pH. The pH of wastewater is considered to be among the most critical factors for the stable operation of sewage plants because it affects the strength and surface charge of the sludge flocs [45]. It can be concluded that the bioaugmentation of the AS with *Pseudomonas* strains made it possible to maintain the system stability because these bacteria positively affected the microbial biodegradation of the contaminants in the sewage and thus helped to maintain stable pH conditions. This observation may indirectly lead to the conclusion that there were no antagonistic relationships between inoculated strains as well as between the exogenous bacteria and autochthonous microorganisms in the AS. Throughout this study, taking into account the removal efficiencies of phenols, the bioaugmented SBRs can be ordered as follows: $L_{KB3} > L_{OR45a+KB4} = L_{OR45a}$. It is also worth emphasizing that the bioaugmentation of the AS with *P. putida* KB3 was slightly more efficient in terms of the utilization of contaminants in the leachate (95%) than the inoculation of the AS with *P. putida* OR45 (90%) as well as with mixed consortium of these bacteria (90%). In turn, autochthonous microorganisms in the AS were not able to completely remove phenolic compounds present in the leachate, in particular 4-methylphenol, 3,4-dimethylphenol, and 2,3,5-trimethylphenol. Moreover, their degradative potential towards phenolic compounds decreased along with the increase in the leachate concentration.

The landfill leachate is often characterized by the high strengths of ammonia nitrogen, which toxicity may inhibit the growth and activity of microorganisms in the AS and thus suppress the biodegradation process [46]. As previously described, *P. putida* OR45a and *P. putida* KB3 withstood high concentrations of the ammonia up to 600 mg/L and 900 mg/L, respectively [26]. Herein, the removal of N–NH$_3$ (88–91%) was much more improved in the bioreactors $L_{OR45a}$, $L_{KB3}$, and $L_{OR45a+KB3}$ as compared to the noninoculated bioreactor L (64%). These results also confirmed our previous assumptions that *P. putida* KB3 may exhibit a higher ability to remove the N–NH$_3$ than *P. putida* OR45a. Similar to the removal efficiency of the COD and phenolic compounds, no synergy between bacteria inoculated into the AS fed with the leachate towards the removal of N–NH$_3$ was observed. The potential of other bacteria *Bacillus cereus* Jlu and *Enterococcus casseliflavus* Jlu in the bioaugmentation of wastewater spiked with the mature landfill leachate originated from the municipal landfill site located near Chang Chun city (China) was assessed by Yu et al. [19]. Their results showed that the removal efficiencies of COD achieved 86% and 90% in effluents inoculated with *B. cereus* Jlu and *E. casseliflavus* Jlu, respectively. In turn, the removal efficiency of N–NH$_3$ was comparable in both cases and reached

about 50%. By comparison, the autochthonous bacteria were able to remove 70% of the COD and 40% of the N–NH$_3$ [19]. Successful bioaugmentation of the landfill leachate from the pond at the Bukit Beruntung landfill (Malaysia) with the *Bacillus salmalaya* 139SI strain was also demonstrated by Dadrasnia et al. [47], who observed the decrease in the COD and N–NH$_3$ by 91% and 80%, respectively, as compared to the nonbioaugmented leachate where removal efficiency of both contaminants was slightly above 30%.

The SOUR is a promising indicator used for characteristics of the AS because it describes the respiration rate of microorganisms, which is strongly related to the microbial activity and vitality [48,49]. Therefore, it can be applied for the control and identification of potential instabilities of the AS systems [50,51]. In this study, the disruption of the performance of nonbioaugmented system spiked with the leachate was observed by the gradual decrease in the SOUR for the AS by 53% along with the increase in the leachate concentration in wastewater. By comparison, the values of this parameter were almost 2–3 times higher for the bioaugmented SBRs fed with the leachate as well as for the inoculated control bioreactors. A similar phenomenon was observed by Eichner et al. [52] who found that microorganisms in the noninoculated control bioreactor collapsed after the shock load of contaminants, which resulted in the oxygen uptake decrease. In research by Contreras et al. [53], the oxygen uptake rate was used for the evaluation of phenol toxicity towards nonacclimated and acclimated autochthonous microorganisms in the AS. It was stated that phenol was toxic to microorganisms, which were not previously adapted to its presence in wastewater, which resulted in a decrease in the microbial respiration by 75% as the phenol concentration increased. Contrary, the respiration of acclimated sludge microorganisms increased within the tested phenol concentrations.

The activity of microorganisms in the AS is extremely important for the appropriate functioning and stability of this ecosystem. However, this parameter is hardly ever considered during the evaluation of the effect of the leachate on the sludge condition. Herein, the impact of the leachate on the activities of dehydrogenase, nonspecific esterase, and catalase was evaluated, as these enzymes play a key role in the metabolism of contaminants in wastewater. The results of the respiratory activity of the sludge were consistent with the outcomes of the microbial enzymatic activity in the SBRs. It was found that the presence of leachate in wastewater affected most the activity of enzymes in the nonbioaugmented bioreactor contributing to the reduction in the activity of dehydrogenases, nonspecific esterases and catalase by 76%, 59%, and 18%, respectively. It can be concluded that the dehydrogenases were the most sensitive enzymes to the toxic components in the leachate. A similar phenomenon was reported by Yao et al. [54] who stated that among five enzymes measured in the AS contaminated with tetrahydrofuran, the activity of dehydrogenases was close to detection level limits and was nearly completely inhibited. This observation was correlated with a significant decrease in the diversity of the AS microbial communities.

The success of bioaugmentation depends, to a large degree, on the ability of the introduced bacteria to survive and display their activities in the AS environment. Our study showed that the enzymatic activity in the bioaugmented AS was higher during the leachate wastewater treatment than in the nonbioaugmented system. The increase in the microbial activity in the inoculated SBRs loaded with the leachate correlated with the increase in the sludge biomass as well as the number of heterotrophic microorganisms, while the gradual reduction in these parameters was observed in the nonbioaugmented bioreactor. It can be concluded that the compounds present in the leachate may not be used for the maintenance and growth of autochthonous microorganisms in the AS but they can be utilized by the introduced *Pseudomonas* strains. This could be indicated by the higher increase in the growth of phenol-degrading bacteria in the bioaugmented AS. The positive correlation between the sludge biomass increase and the efficiency of contaminants degradation in the bioaugmented AS was also observed by Yao et al. [55], who found that tetrahydrofuran was completely removed from wastewater when the sludge biomass increased from 2.1 g/L to 7.3 g/L.

The high metabolic capacity of the AS microbiome is considered to be extremely important for efficient wastewater treatment in sewage plants [56]. It was found that microorganisms in the SBRs fed

with leachate were less metabolically active than bacteria in the control bioreactors. Nevertheless, the addition of *Pseudomonas* strains into the SBRs allowed to maintain higher microbial activity than in the bioreactor filled with noninoculated biomass, which contributed to the increase in functional diversity in the bioaugmented systems. Throughout this study, the microbial activity and functional capacity in the bioaugmented SBRs can be ordered as follows: $L_{OR45a+KB3} > L_{OR45a} = L_{KB3}$. The addition of exogenous bacteria into the AS is expected to lead to changes in microbial communities [57]. As described by Fang et al. [58] some microorganisms introduced into the bioreactor might gain constant proliferation, exerting a key role in the removal of pollutants. Here, the low functional diversity of 1.66 was correlated with a lack of ability of microbial communities to utilize specific compounds, in particular, the phosphorus sources. It was found that the occurrence of the leachate in wastewater led to the metabolic specialization of the AS microorganisms. It can be concluded that the changes in the utilization patterns of carbon sources for the bioaugmented SBRs can be the result of the *Pseudomonas* strains persistence in the AS, which is evidenced by the increase in the utilization capacity of phenolic compounds and carboxylic acids in these systems.

The novelty of this work was mainly related to the use of the active biomass and stress biomass indices to assess the impact of the leachate as well as bioaugmentation on the AS. The study performed by Archibald et al. [59] revealed that the ATP measurements provided useful monitoring of the proportion of viable cells and were toxicity indicators in the AS process. The research by Pistelok et al. [42] showed that the ATP analyses helped in the quick assessment of wastewater plant effectiveness and could complement other indicators e.g., organic carbon, total suspended solids, biochemical and chemical oxygen demand. Herein, the presence of the leachate in wastewater constituted a strong stress factor for microorganisms in the AS. It was found that the bioreactor did not work properly when the biomass stress index was higher than 50%. Our results indicated that the addition of exogenous bacteria into the AS alleviated the toxic effect of the leachate and thus reduced the stress in this environment.

The condition of the AS in the tested SBRs was evaluated by the measurement of the sludge volume index. It is worth emphasizing that the bioaugmentation improved the settling properties and quality of the AS in the SBRs. This can be explained by good coaggregation abilities exhibited by the bioaugmentation candidates [26]. The improvement of the AS settling properties via the bioaugmentation with *Halobacter halobium* was also described by Kargi et al. [60]. On the other hand, the low value of the sludge volume index for the bioreactor L could indicate a considerable dispersion of the AS flocs, which could be correlated with the increase in the turbidity of supernatant liquid.

Multivariate analysis performed in this work allowed to establish important relationships between the bioaugmentation and operational parameters of the wastewater treatment process in the AS. Although it was indicated that the presence of the *Pseudomonas* strains was responsible for the alleviating of the variations in the AS, the inoculation of the bioreactor with *P. putida* KB3 as well as with mixed microbial consortium stimulated many more processes in the SBRs, e.g., the increase in microbial enzymatic activity, MLSS and COD removal efficiency. On the other hand, the introduction of *P. putida* OR45a into the bioreactor contributed to the increase in the utilization of phosphorus sources. Combining physicochemical and biological analyses allowed us to identify the general effect of the bioaugmentation on the AS condition and SBRs performance.

## 5. Conclusions

Herein, we explored the bioaugmentation potential of two newly isolated *P. putida* OR45a and *P. putida* KB3 for enhancing the biodegradability of the phenolic landfill leachate in the AS. Our results showed that both strains have the potential to improve the efficiency of the conventional biological treatment of the phenolic landfill leachate and can alleviate the toxic impact of the leachate-spiked effluent on autochthonous microorganisms. However, it should be noted that the inoculation of the AS with *P. putida* KB3 was more effective in the improvement of the bioreactor operation than its bioaugmentation with *P. putida* OR45a and bacterial consortium. This was manifested in the significant improvement of the removal efficiency of phenolic compounds, COD, and $N–NH_3$ and contributed

to the increase in microbial enzymatic activity and generation of better-quality effluent. Therefore, *P. putida* KB3 strain can be considered as having the best potential to improve the efficiency of the conventional sewage plant dealing with the phenolic landfill leachate. Our research also emphasizes the successful combination of standard operational parameters with simple and more accurate methods (e.g., analyses of the ATP concentration, microbial metabolic activity, and functional potential) instead of focusing on the expensive genetic analyses of microbial community structure, as they provide early warning of the performance status of the AS and sewage plant.

**Supplementary Materials:** The following are available online at http://www.mdpi.com/2073-4441/12/3/906/s1, Table S1: The changes in the removal efficiency of the COD and N-NH3, pH and SOUR values in the non-inoculated and bioaugmented SBRs during 96 days of the wastewater treatment; Table S2: The changes in the concentration of phenolic compounds and wastewater quality index in the non-inoculated and bioaugmented SBRs fed with the synthetic wastewater and phenolic landfill leachate during 96 days; Table S3: The values of functional capacity indices and kinetic parameters ($A_{max}$ and $Avt_{1/2}$) for the non-inoculated and bioaugmented AS, fed only with the synthetic wastewater and cotreated with the phenolic landfill leachate during 96 days; Table S4: The values of carbon, nitrogen and phosphorus usage indices for the non-inoculated and bioaugmented AS, fed only with the synthetic wastewater and cotreated with the phenolic landfill leachate during 96 days.

**Author Contributions:** Conceptualization, J.M. and A.M.; formal analysis, J.M., A.P., and J.Ż.; data curation, J.M.; writing—original draft preparation, J.M. and A.M.; writing—review and editing, A.M.; visualization, J.M.; supervision, A.M.; project administration, J.M. and A.M.; funding acquisition, J.M and A.M. All authors have read and agreed to the published version of the manuscript.

**Funding:** The paper was prepared in connection with the work done under the project granted based on decision DEC-2016/23/N/NZ9/00158 and financed by the National Science Centre (Poland).

**Acknowledgments:** The research was performed due to the courtesy of the Chorzowsko-Świętochłowickie Municipal Water and Sewage Company LLC and the Head of Wastewater Treatment Plant Piotr Banaszek. The authors are also very grateful to Izabela Greń for helping with the analysis of phenolic compounds in wastewater and Katarzyna Malarz for helping with the analysis of ATP in the activated sludge.

**Conflicts of Interest:** The authors declare that the research was conducted in the absence of any commercial or financial relationships that could be construed as a potential conflict of interest.

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
