# Peer review of "Analysis of the Bioaugmentation Potential of Pseudomonas putida OR45a and Pseudomonas putida KB3 in the Sequencing Batch Reactors Fed with the Phenolic Landfill Leachate"

_water, doi:10.3390/w12030906_

Round 1

Reviewer 1 Report

This is a well-written paper. The methods and results are overly described. The current length of the paper (23 pages) would fail to capture the attention of the readers. So many subsections are confusing. Unnecessary Tables (2 - 5) could be removed or greatly simplified. Other than that, the paper has the merit to be accepted in Water. 

I am suggesting to greatly reduce the length of this paper (by at least 40%) for better readability. 

Author Response

Dear Reviewer,

Thank you very much for your work and valuable comments. We have corrected the manuscript according to your suggestions. All changes and explanations are listed in an enclosed table.

We will be very grateful if you accept our corrections.

Sincerely

Justyna Michalska and other co-authors

Reviewer 2 Report

General comments

In this study, four SBR reactors were bioaugmented with and without strains OR45a, KB3, and a mixture of them to investigate their performance for landfill leachate polluted with phenolic compounds. The other four SBR reactors as controls were also simultaneously operated for comparison. The performance of all reactor was analyzed in detail through various indices and microbial community. The data deserves to be published. However, the manuscript should be thoroughly revised before publication especially in language presentation.

Specific comment

- The current title is quite awkward. This study aimed to treat landfill leachate polluted with phenolic compounds. It is not natural to say that “reactors contaminated with…”. Please revise the title.

- Terms “contaminated reactors”, “contaminated AS”, “polluted AS”, and “polluted reactors” were used throughout the manuscript. What do they mean? It is very ambiguous. Landfill leachate was actively fed to reactors with an objective of treatment. It is inappropriate to use these expressions.

- Too many abbreviations were used throughout the manuscript. Of course, the abbreviation is recommended for some terms repeated regularly and only in necessary cases. However, abbreviation abuse would prevent readers from understanding.

- The conclusion should be revised to prevent a general conclusion. Specific information for this study should be provided. Four different cases (without bioaugmentation, with strain OR45a, KB3, and mixture) were investigated, so at least which reactor gave the best performance, what is different from those cases?

Author Response

(The authors gave the same response as above.)

Reviewer 3 Report

This is a novel, relevant study that I enjoyed reviewing. Overall, my comments are minor:

  1. There were a few minor grammatical and citation errors. For instance, line 52 states "referred as bioaugmentation" instead of "referred to as bioaugmentation". On line 87, the sentence reads "in accordance with 28 and 29"- it would be better to state the authors' last names or state it's in accordance with previous studies. 
  2. I don't believe Klimzowiec needs quotations
  3. Figure 1 is helpful is explaining the experimental set-up, but it seems a bit confusing regarding if the sedimentation and filling are being cycled back in the experimental groups and controls, respectively. 
  4. Section 2.11., which discusses the statistics, could be made more concise. For instance, it mentions that Excel, ScyPy, R, and Prism were used for statistics but it does not describe why all of these programs were used nor which statistical analysis was conducted in each program. In addition, it does not appear that the PCA or ANOVA results were included, but may be helpful in reproducing findings.
  5. Figure 3 provides a lot of information and it's currently a bit difficult to clearly see everything, particularly the labels and asterisks. I would recommend trying to increase the font size or breaking up the figures in two separate figures. In addition, I would add information regarding statistical significance in the figure 3 caption. Figure 5 also have very small font that is difficult to read. 
  6. Check for re-defining acronyms, such as on Line 608 where SOUR was re-introduced. 

For some additional positives, this researchers did not use expensive genetic techniques and kept a practical prospective to the project-- increasing the relevance and chances for full-scale implementation. In addition, the figures are very nice and clearly present data.

Author Response

(The authors gave the same response as above.)

Round 2

Reviewer 2 Report

The paper was significantly improved and can be accepted for publication.